# Comparative metabolism of conjugated and unconjugated pterins in *Crithidia*, *Leishmania* and African trypanosomes

**Han B. Ong, Susan Wyllie, Alan H. Fairlamb** ⓘ *

Division of Biological Chemistry & Drug Discovery, School of Life Sciences, University of Dundee, Dundee, United Kingdom

* a.h.fairlamb@dundee.ac.uk

## Abstract

Mammalian cells synthesise tetrahydrobiopterin *de novo*, an essential cofactor for hydroxylation of aromatic amino acids, cleavage of ether lipids and the synthesis of nitric oxide. In contrast, kinetoplastid parasites are pterin auxotrophs and none of the above metabolic functions can account for the essential requirement of an unconjugated pterin for growth. Here we investigate the pterin requirements for growth and survival of two medically important parasites (*T. brucei* and *L. major*) in comparison with the model insect parasite, *Crithidia fasciculata*. The pterin concentration required to support 50% of maximum growth of each parasite was determined in defined pterin-free media for a variety of naturally occurring pterins. *T. brucei* and *C. fasciculata* showed an identical order of preference with the most active being 6-biopterin, followed by dihydrobiopterin > tetrahydrobiopterin > L-neopterin > sepiapterin. In contrast, *L. major* showed a pronounced growth preference (>200-fold) for the reduced pterins over the the oxidised forms 6-biopterin and L-neopterin. The unnatural isomers 7-biopterin or D-neopterin supported growth poorly, or not at all, in these organisms. Other pterins were inactive. HPLC analysis of pterins supporting growth established that these were metabolised to the tetrahydro-forms (>95%) with no evidence of further interconversion. In the absence of pterins, the parasites failed to grow and lost viability with <1% surviving beyond 5–14 days. Relatively high concentrations of folate or dihydrofolate (>500 nM) could support growth in the absence of unconjugated pterin and HPLC analysis identified pteridoxamine and 6-hydroxymethylpterin (as tetrahydro-form) in cell extracts. A common feature of pterins that support growth is the presence of at least one or more linear carbon substituents at position 6 of the pteridine ring with at least one hydroxyl group, ideally in the 1*S* configuration. The possible essential roles of these important metabolites are discussed.

**Data availability statement:** All relevant data are within the manuscript and its Supporting Information files.

**Funding:** This work was supported by grants from the Wellcome Trust [079838], [083481] and [203134] to AHF. The funders had no role in study design, data collection and analysis, decision to publish, or preparation of the manuscript.

**Competing interests:** The authors have declared that no competing interests exist.

## Author summary

Over 75 years have passed since the discovery of " Crithidia factor" – biopterin – as an essential cofactor involved in several important metabolic functions in mammalian cells. Despite established roles of biopterin as a cofactor in the hydroxylation of aromatic amino acids, the cleavage of glyceryl ether lipids and in the synthesis of nitric oxide in mammalian cells, these metabolic functions either do not exist in kinetoplastids or are unable to account for the essential growth requirements for pterins in these parasites. Here, we have determined structural requirements of pterin analogues that can support growth and survival of three important genera in the order Kinetoplastida. Some quantitative differences are noted in pterin preferences between *C. fasciculata* and *T. brucei*, and those of *L. major*. We find that the sole fate of intracellular pterins in these parasites is reduction to the corresponding tetrahydro intermediate. Folic acid can spare the growth requirement for an unconjugated pterin possibly due to trace amounts being converted to 6-hydroxymethylpterin. Future research directions for elucidation of the essential functions of pterins in these trypanosomatid parasites are discussed.

## Introduction

Pterins are a class of nitrogen-containing heterocyclic compounds that share a characteristic 2-amino-4-oxopteridine moiety and can be further classified as conjugated or unconjugated [1,2]. These compounds can be reduced at the 5, 6, 7 and 8 positions of the pteridine ring to exist in different oxidation states (Fig 1A). Conjugated pterins contain a *p*-aminobenzoate substituent at the 6-position of the pteridine moiety coupled a single glutamate residue and can be further polyglutamated to facilitate retention within cells (Fig 1B). Unconjugated pterins contain neither of these substitutions and instead have varying lengths of hydrocarbon side chains on the 6 or 7-position of the moiety (Fig 1C). Despite their structural similarities, both classes of pterins are required for many important metabolic processes. Conjugated pterins are more commonly referred to as folates whose reduced derivatives are essential cofactors for several enzymes catalysing one-carbon transfer reactions [3–5]. The term "pterin" is used primarily to describe unconjugated pterins. Oxidised pterins, such as xanthopterin and leucopterin, are major components of butterfly wings [6], while the fully reduced tetrahydro-form of biopterin (tetrahydrobiopterin, $H_4B$) is an essential cofactor for various one-electron transfer enzymatic reactions [7,8]. Pterins are also essential metabolic intermediates in the biosynthesis of a 7-deazaguanine derivative, queuosine, in eubacteria. Queuosine is incorporated into the wobble position 34 of the anticodon loop of tyrosyl, asparaginyl, aspartyl and histidinyl tRNAs not only in eubacteria, but also by salvage in plants, animals and fungi [9]. This modification is important for translational fidelity [10,11]. Like other eukaryotes, trypanosomes do not appear to be able to synthesise queuine de novo and obtain it from their environment

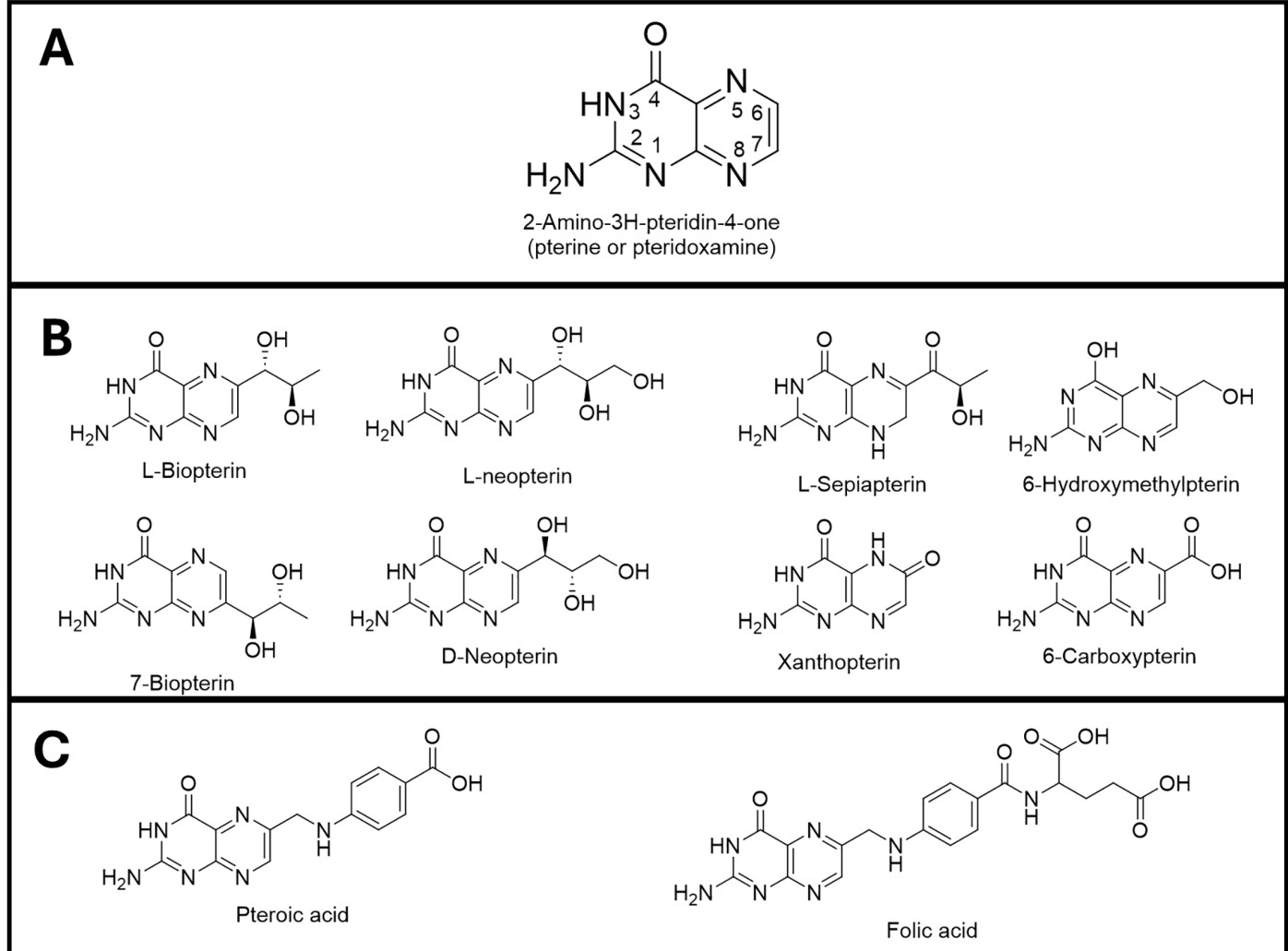

**Fig 1. Chemical structure of unconjugated and conjugated pterins.** Panel A: Basic pterin moiety and the numbering system in the pteridine ring. Panel B: Chemical structures of different unconjugated pterins used in the present study. Panel C: Chemical structures of pteroic acid and folic acid which are pterins conjugated to an aminobenzoate group with and without glutamylation, respectively.

[12,13]. Although levels of queuosine in tRNA can change in response to availability of certain amino acids, these changes have negligible effect on growth rate or global protein synthesis, eliminating this as an essential role for pterins [12,13].

$H_4B$ is the best studied pterin due to its important biological functions [14]. In mammalian cells, $H_4B$ is synthesised *de novo* from guanosine triphosphate (GTP) and acts as an essential cofactor for several enzymatic reactions such as the hydroxylation of aromatic amino acids (phenylalanine, tyrosine and tryptophan) by their respective hydroxylases [15–17]. $H_4B$ is also required by nitric oxide synthases (NOS) to produce nitric oxide, both an important signalling molecule and a powerful antimicrobial agent [18–20]. In lipid metabolism, the enzyme glyceryl-ether monooxygenase also utilises $H_4B$ for the hydroxylation of glyceryl-ethers, resulting in their subsequent cleavage [21,22].

*Leishmania spp.* and *Trypanosoma ssp.* are protozoan parasites that cause some of the most neglected tropical diseases known as the Leishmaniases, and the South American and African trypanosomiases. *Crithidia spp.* are

parasites of insects that have been used as convenient models for the study of these fastidious vertebrate parasites [23,24]. Unlike mammalian cells, trypanosomatids such as *Leishmania spp.* and *Crithidia spp.* are unable to synthesise pterins *de novo* and are therefore pteridine auxotrophs. The importance of pterins was first described in the 1950s when biopterin was found to be an important growth factor for the insect parasite, *Crithidia fasciculata* [25,26]. The term "*Crithidia* growth factor" originated from these studies and is used to describe pterins, including biopterin, that are able promote the growth of *C. fasciculata*. A fully defined media (KD media) was formulated for culturing of *C. fasciculata* where high levels of folate supplementation were determined to be required for cell growth [27]. This requirement for high folate was significantly lowered by the addition of trace amounts of biopterin [13] and this observation has been used in the biological assay of unconjugated pteridines in biological fluids and tissues [27,28]. Despite the extensive knowledge of the roles of pterins in other organisms, the function(s) of pterins within the Kinetoplastida is still not yet understood.

Pterin metabolism in *Leishmania spp.* was studied extensively following the discovery of an enzyme, pteridine reductase 1 (PTR1), which plays a role in parasite resistance to the classical antifolate drug methotrexate [29,30]. This enzyme plays a pivotal role in the generation of $H_4B$ from salvaged biopterin in trypanosomatids [31,32]. PTR1 from both *Leishmania major* and *Trypanosoma brucei* has also been shown to be capable of regenerating $H_4B$ from *quinonoid* dihydrobiopterin ($qH_2B$), an intermediary product of pterin-dependent hydroxylation reactions [2]. This reaction is normally catalysed by a dedicated NADH-dependent *quinonoid* dihydropteridine reductase ($q$DPR) that is also present in *L. major* but absent in *T. brucei* [2,33]. The essentiality of PTR1 in *T. brucei* was confirmed by the knockdown of PTR1 by RNAi, which, unlike *L. major* PTR1 knockout mutants, could not be rescued by any pterin supplementation [34]. These findings made PTR1 an attractive novel target for drug discovery initiatives against *T. brucei* for the treatment of Human African Trypanosomiasis (HAT). However, inhibitors designed against this enzyme have been met by a lack of cellular potency [35]. This study is aimed at better understanding the requirement for pterins in the Kinetoplastida, particularly *T. brucei*. Specifically, we investigate the growth requirements and metabolism of pterins in *T. brucei* in a comparison to *L. major* and *C. fasciculata*. Our findings should facilitate future drug development against this area of parasite metabolism.

## Methods

*Organisms and reagents* – All pterins were purchased from Schircks laboratories. Other chemicals and reagents used in this study were of the highest grade and purity available. *C. fasciculata* clone HS6 [36] was cultured at 28 °C with shaking at 200 rpm in medium D plus 2.27 nM folate [27]. *T. brucei* bloodstream-form S427 was cultured at 37 °C in a modified HMI-9 medium containing 31 nM folate (FDM [37]) supplemented with 10% dextran/charcoal-treated foetal calf serum (FCS, Hyclone) and 15 µg ml$^{-1}$ gentamycin sulphate (G418, Invitrogen). *L. major* promastigotes (Friedlin strain; WHO designation MHOM/JL/81/Friedlin) were cultured at 28 °C with shaking at 150 rpm in M199 medium (Invitrogen, folate content 22.7 nM) containing 0.5% FCS (PAA). L-Biopterin (10 µM) was added for routine culturing of all parasites.

*Concentration of pterins required for half maximal growth* – parasites were depleted of pterins by subculturing (1 x 10$^4$ cells ml$^{-1}$) in pterin-free media for 72 h. The concentration of pterins required for growth were determined in 96-well microtiter plates in a final culture volume of 200 µl per well and an initial parasite seeding density of 1 x 10$^4$ cells ml$^{-1}$. The pH of the assay media was adjusted to correct for HCl (1 mM, $H_4B$) or NaOH (1 mM, all other pterins) used for the preparation of stock pterin solutions. Cells were incubated at their normal culture media as above. Cell densities were determined daily using a CASY cell-counter. To assay cell viability of parasites grown in the absence of pterins, cells were subcultured in fresh medium for 72 h before growth was determined using a resazurin-based assay [38,39]. The ability of each pterin to support 50% maximum cell growth ($GC_{50}$) was determined by comparison with cultures supplemented with 10 µM L-biopterin (positive control for maximum growth, 100%). $GC_{50}$ values were determined by plotting cell growth with respect to pterin concentration and analysed by non-linear regression with a 4-parameter logistic equation in GraFit.

$$y = \frac{Range}{1 + \left(\frac{x}{GC_{50}}\right)^s} + Background$$

*Analysis of intracellular pterins* – Intracellular pterins of parasites were determined by HPLC as previously described [2]. Intracellular pterin concentrations were calculated based upon previously published cell volumes of 10.5, 5.8 and 4.3 µl per $10^8$ cells for *C. fasciculata* [40], *T. brucei* [41], and *L. major* [42], respectively.

 *Enzymatic assays* – Cell lysates were prepared as previously described [43]. Trypanothione reductase and PTR1 activities in these lysates were determined as previously described [44,45].

## Results and discussion

*Effect of extracellular L-biopterin concentrations on cellular pterin levels in C. fasciculata* – The insect parasite, *C. fasciculata* was used as a model organism to develop methods for studying pterin metabolism in other trypanosomatids. Unlike *T. brucei*, these parasites can be cultured in a fully defined medium that does not require serum, thus enabling controlled modulation of pterin structure and concentration. Initially, *C. fasciculata* was grown in medium D supplemented with 4 nM L-biopterin, as recommended by Dewey and Kidder [27]. However, under these conditions, intracellular pterin content was below the limits of detection in our HPLC assay. This problem was resolved by increasing the biopterin concentration to 50 nM in the culture medium. Under these conditions, biopterin was readily detected in log-phase cells (harvested by centrifugation at ~1 x $10^7$ cells ml$^{-1}$) such that total biopterin was proportional to cell numbers between $1 \times 10^7$ and $2.5 \times 10^8$ per assay (Fig 2). All subsequent HPLC analyses were carried out on ~5 x $10^7$ cells. Biopterin was found to be the predominant intracellular pterin in these log-phase parasites, with the fully reduced H$_4$B constituting more than 95% of the total (Fig 3). The total intracellular content of biopterin in mid-log phase *C. fasciculata* cultured in 50 nM biopterin (Fig 2) was determined as $62.2 \pm 0.4$ pmol/ $10^8$ cells, from which an intracellular concentration of $5.9 \pm 0.04$ µM can be calculated.

 Biopterin is essential for growth of *L. major* with decreases in H$_4$B levels regulating parasite differentiation from the sand fly promastigote stage to the infective mammalian metacyclic stage [47]. A similar decrease in H$_4$B concentration was also found with *C. fasciculata* when cultured in 0.1 µM biopterin (Fig 3, pink). Mid-log phase cells (harvested at ~1 x $10^7$ cells ml$^{-1}$) were found to maintain the highest concentrations of biopterin ($9.6 \pm 0.03$ µM), with levels decreasing 6-fold as cells entered stationary phase. The degradation of PTR1 during stationary phase has been proposed to be directly responsible for the decrease in H$_4$B levels in *Leishmania* [48]. However, this was found not to be the case with *C. fasciculata*, where PTR1 activity in clarified lysates of these parasites was maintained at $55.9 \pm 2.4$ pmol min$^{-1}$ mg$^{-1}$ (~0.02% of the trypanothione reductase activity, $330 \pm 10.3$ nmol min$^{-1}$ mg$^{-1}$, used as a control for cell breakage) throughout the different growth phases. Instead, HPLC analyses of media supernatants from stationary-phase cultures indicated that the complete exhaustion of L-biopterin was a more credible reason for the decrease in pterin levels. To test this hypothesis, pterin levels were measured in cells cultured in the presence of saturating amounts of L-biopterin (10 µM) during different growth stages (Fig 3, blue). Consistent with the previous observations, H$_4$B (>95%) was again the predominant form of intracellular biopterin in these cells. However, total biopterin levels ($21.8 \pm 0.13$ µM) were considerably higher compared to cells cultured in 0.1 µM L-biopterin, across the different growth stages. Of note, H$_4$B levels remained high even in stationary phase parasites. These results suggest the intracellular pterin levels in *C. fasciculata* are directly dependent upon the availability of pterin in the media and that these parasites can accumulate pterins against a concentration gradient.

 *Biopterin requirement in the trypanosomatids* – The doubling time of *C. fasciculata* cultured in medium D supplemented with 10 µM L-biopterin was determined to be $6.57 \pm 0.03$ h (Fig 4A, open symbols). Cells sub-cultured in biopterin-free medium grew normally for 72 h before growth ceased by 120 h (Fig 4A, closed symbols). Biopterin-starved parasites were present for up to 14 days in subcultures, although the parasites progressively lost viability commencing on day 4 (Fig 4A,

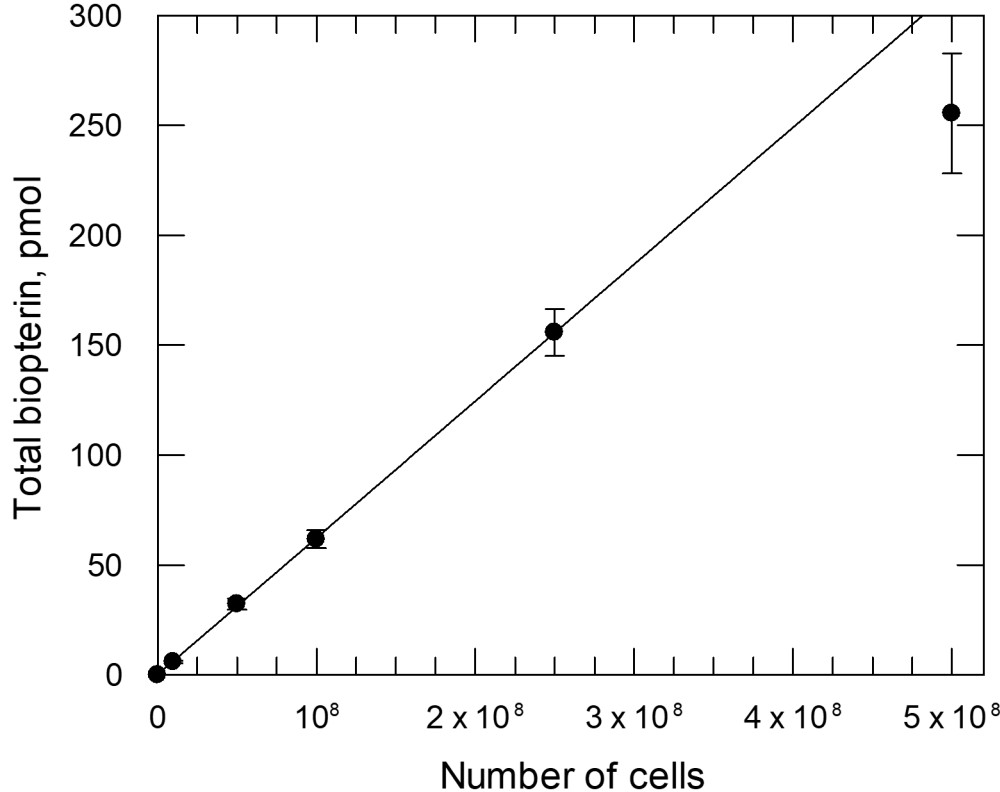

**Fig 2. Linearity of HPLC based assay for intracellular pterin analysis.** *C. fasciculata* were treated with iodine in 0.1 M HCl to oxidise 7,8-dihydrobiopterin and tetrahydrobiopterin to biopterin [46], protein removed by centrifugation and the supernatant analysed by HPLC [2]. Total biopterin was proportional to cell numbers up to $2.5 \times 10^8$ cells (linear regression coefficient of 1.00).

inset). These observations are consistent with pterin levels in cells falling below the limit of detection after 72 h subculturing in biopterin-free medium D.

The importance of nutritional supplementation of biopterin was further demonstrated in *T. brucei*. In the first instance, bloodstream trypanosomes were conditioned to grow in minimal medium where growth is only permissive when L-biopterin is included. *T. brucei* was cultured in a previously described folate-deficient medium [34], with the "10% FCS and 0.5% Serum-plus" replaced by 10% charcoal/dextran-treated FCS (FDM$^{dex}$). The medium was supplemented with 10 µM L-biopterin and the doubling times for these parasites were determined to be $8.4 \pm 0.08$ h (Fig 4B). Consistent with observations made with *C. fasciculata*, these parasites continued to grow normally for about 3 days when cultured in the absence of biopterin before growth ceased (Fig 4B). At this point the intracellular pterin levels were below the limit of detection. However, unlike *C. fasciculata*, despite supplementation with fresh biopterin-free media, these cells were unable to survive in culture beyond day 4, with prominent loss of viability evident by day 5 (Fig 4B, inset).

Growth of *L. major* promastigotes in biopterin-replete medium replicated that observed with *C. fasciculata* under the same conditions (Fig 4C). Linear growth was observed when subcultured in replete media with a doubling time of $10.4 \pm 0.08$ h. In biopterin-free medium, growth ceased entirely by day 3. Progressive loss of viability over time was also observed.

*Comparing the ability of different pterins to support growth - C. fasciculata* can utilise folic acid, in place of L-biopterin for optimal growth in medium D (Fig 5A). However, the difference in pterin concentration required to support 50% of maximum cell growth (GC$_{50}$) spans 4 orders of magnitude ($0.2 \pm 0.01$ nM versus $3,680 \pm 190$ nM for biopterin and folate,

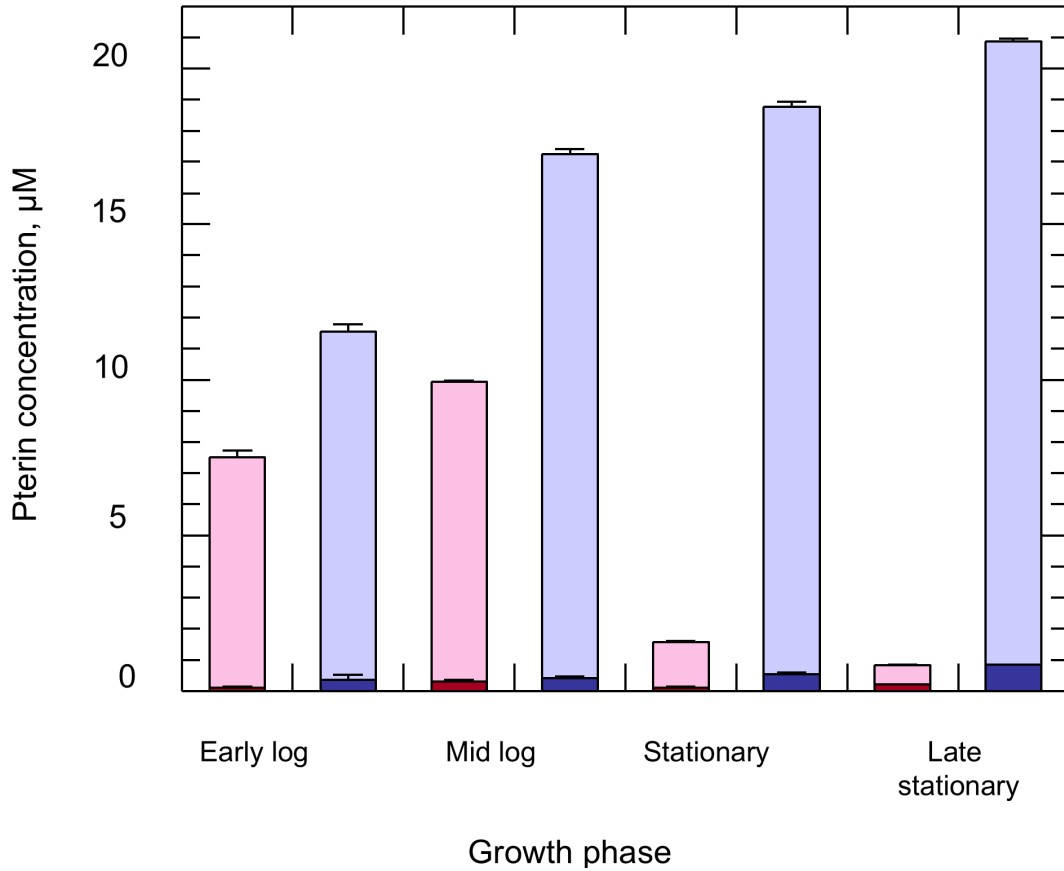

**Fig 3. Intracellular pterin content of *C. fasciculata* during growth.** *C. fasciculata* was cultured in KD medium supplemented with either 100 nM (pink) or 10 µM (blue) of L-biopterin. Cells were cultured at an initial seeding density of 5 x $10^4$ cells ml$^{-1}$ and harvested during different growth stages namely early log (1 x $10^6$ cells ml$^{-1}$), mid-log (1 x $10^7$ cells ml$^{-1}$), early stationary (5 x $10^7$ cells ml$^{-1}$) and late stationary (5 x $10^7$ cells ml$^{-1}$) phases, oxidised with iodine and analysed by HPLC for $H_4B$ (light bars) and biopterin plus $H_2B$ (dark bars). Results are presented as means±S.D. of triplicate measurements.

respectively). *T. brucei* has a similar GC$_{50}$ of 0.83±0.11 nM for biopterin to that determined for *C. fasciculata.* However, only 30% of maximum growth could be achieved with folate up to 5 µM (Fig 5B). *L. major* required considerably more biopterin to support growth (GC$_{50}$ = 567±31 nM) and folate was incapable of supporting growth in the absence of biopterin up to the maximum concentration tested (500 µM) (Fig 5C).

  *C. fasciculata* can use a range of pterins in place of biopterin to support growth [27]. To determine whether this is a generic feature amongst the trypanosomatids, a selection of pterins were tested for their ability to restore growth in pterin-depleted parasites. Several pterins were found to support growth at optimal concentrations (Table 1), while becoming inhibitory at higher concentrations (Table 2). L-Biopterin and its reduced forms were the most adept in supporting the growth of *C. fasciculata*, with GC$_{50}$ values in the sub-nanomolar range. This was followed by L-neopterin and L-sepiapterin with GC$_{50}$ values in the low nanomolar range. D-neopterin and 7-biopterin were considerably less efficient in supporting growth with GC$_{50}$ values that were >800-fold and >8,000-fold higher than their respective isoforms, L-neopterin and L-biopterin. Conjugated pterins (pteroic acid and folic acid) were intermediary in supporting growth with GC$_{50}$ values in the low micromolar range, with the parasites found to prefer $H_2F$ over the fully oxidised folic acid. The sub-nanomolar GC$_{50}$-values for L-biopterin and L-neopterin are in reasonable agreement (within 2- to 9-fold) with the previous work of Dewey and Kidder [27] and Rembold [49]. The general ranking of different pterins' ability to support growth is also in

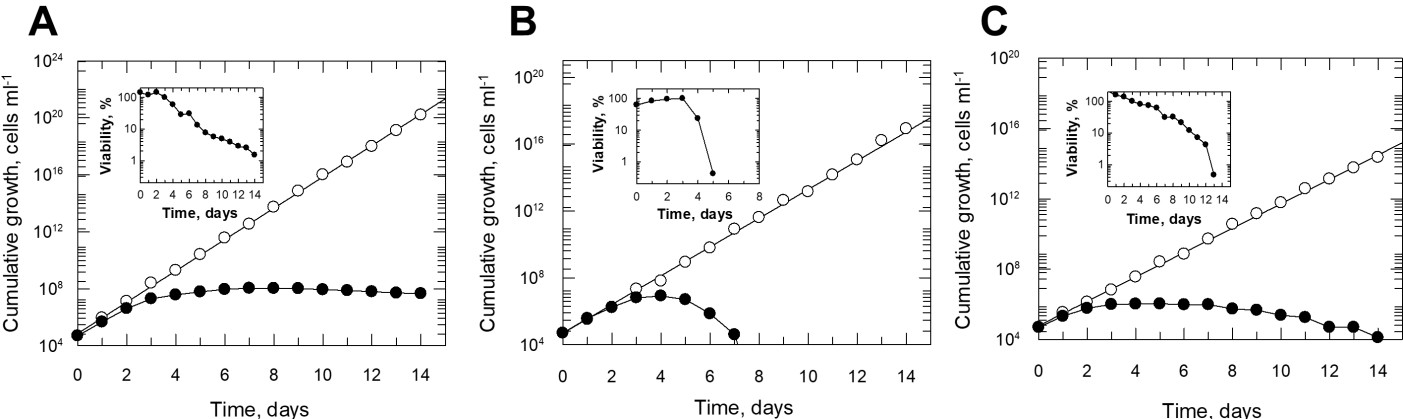

**Fig 4. Growth analyses of *C. fasciculata*, *T. brucei* and *L. major* cultured in media with and without 10μM L-biopterin.** Cumulative growth curves of *C. fasciculata* (panel A), *T. brucei* (panel B) or *L. major* (panel C) cultured in media supplemented with 10 μM L-biopterin (open circles) or without L-biopterin (closed circles). Cell densities were determined daily using a CASY cell-counter. Cells were harvested and resuspended in fresh media with or without L-biopterin every third day. The inset shows the viability of biopterin-free cultures. Viability of biopterin-free cultures was assessed by daily resuspension at a final density of 1 x 10⁴ cells ml⁻¹ in fresh media containing 10 μM L-biopterin. Culture density was measured after a further 72 h incubation and the viability of these cells expressed as a percentage of maximum growth from day 3 cultures. Results are the mean of triplicate measurements.

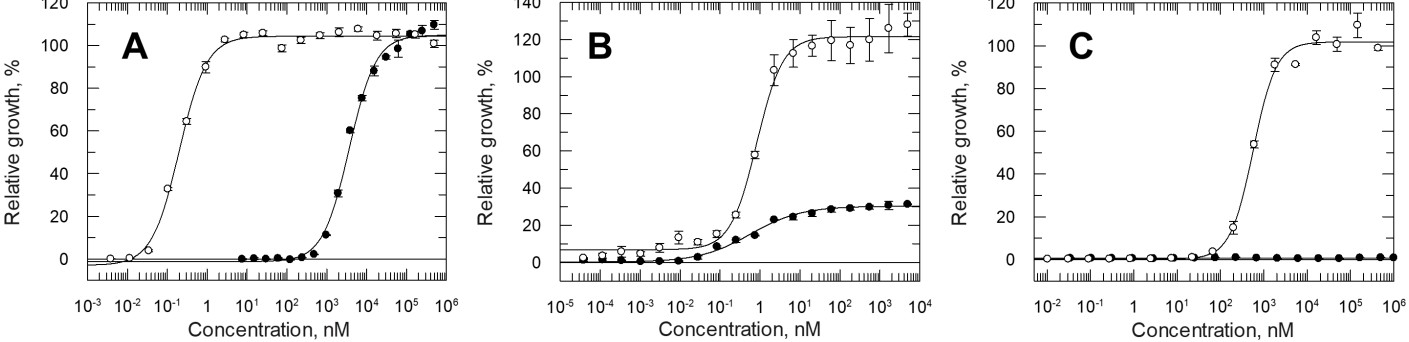

**Fig 5. Growth of parasites cultured in varying concentrations of folic acid and biopterin.** Panels A, B and C respectively show the $GC_{50}$ curves of *C. fasciculata*, *T. brucei* and *L. major* cultured in varying concentrations of biopterin (open circles) and folic acid (closed circles). Pterin-depleted cells were seeded at densities as specified in methods and cell growth was quantified after 72 h using a resazurin-based cell-viability assay. Results were analysed by 4-parameter non-linear regression using GraFit and presented as means ± S.E.M of triplicate measurements.

good agreement. L-Sepiapterin, an intermediate in the salvage and de novo synthetic pathways of tetrahydrobiopterin in humans [50], was only moderately active in our assay compared to Dewey and Kidder (58.3 versus 0.34 nM). This discrepancy may be due to an impurity of the latter material, which was isolated from a *Drosophila melanogaster* mutant in the earlier study [27]. Pteridoxamine, 6-carboxypterin, and xanthopterin were reported as weakly active ($GC_{50}$ > 50 μM) in previous studies [27,49], but were unable to support growth in our experiments. The pterins that support maximal growth at lower concentrations but inhibit growth at higher concentrations (Table 2) are reduced pterins or the unnatural isomers of biopterin or neopterin. The toxicity associated with high concentrations of reduced pterins could be due to autoxidation in the media causing oxidative stress [51] and/or high substrate inhibition of PTR1 [2,32].

*T. brucei* was found to have a similar growth requirement for unconjugated pterins compared to *C. fasciculata*, with an identical ranking order for the top six pterins listed in Table 1, albeit with higher $GC_{50}$ values (4.2- to 12.8- fold higher). In

**Table 1. Pterin requirement for growth in *C. fasciculata*, *T. brucei* and *L. major*.** Concentration required for half maximum growth ($GC_{50}$) were determined as described in the methods. Values are the mean±S.E.M of triplicate measurements.

| Pterin | Maximum concentration tested (µM) | *C. fasciculata* GC50 (nM) | *T. brucei* GC50 (nM) | *L. major* GC50 (nM) |
|---|---|---|---|---|
| L-Biopterin | 500 | 0.20±0.01 | 0.83±0.11 | 565±30 |
| $H_2B$ | 500 | 0.21±0.02 | 1.64±0.29 | 2.85±0.23 |
| $H_4B$ | 1000 | 0.62±0.06 | 7.02±0.95 | 1.02±0.10 |
| L-Neopterin | 500 | 1.02±0.13 | 13.1±1.08 | 13,600±570 |
| L-Sepiapterin | 500 | 58.3±8.4 | 670±12 | 810±17 |
| D-Neopterin | 500 | 830±72 | >50,000[1] | >500,000 |
| 7-Biopterin | 500 | 1,650±169 | >50,000[1] | >500,000 |
| $H_2F$ | 500 | 486±37 | 845±65 | 3,510±1.5 |
| Folic acid | 500 | 4,480±263 | 0.56±0.10[2] | >500,000 |
| Pteroic acid | 250 | 1,740±58 | >250,000[3] | >250,000 |
| Pteridoxamine | 250 | >250,000[3] | >250,000[3] | >250,000 |
| Xanthopterin | 250 | >250,000[3] | >250,000 | >250,000 |
| Isoxanthopterin | 125 | >125,000 | >125,000[3] | >125,000 |
| 6-Carboxypterin | 125 | >125,000[3] | >125,000 | >125,000 |

[1]Growth >40% of positive control; unable to determine actual $GC_{50}$ as growth inhibition results at concentrations above 50 µM

[2]Unable to grow beyond ~30% of positive control despite low $GC_{50}$ value (see Fig 5B).

[3]Marginal growth at concentrations above 30 µM, but <20% of positive control at maximum assay concentrations

**Table 2. Pterins supporting maximum growth at low concentrations that inhibited growth at higher concentrations.**

| Pterin | *C. fasciculata* (µM) | *T. brucei* (µM) | *L. major* (µM) |
|---|---|---|---|
| $H_4B$ | >10 | >10 | >10 |
| $H_2B$ | –[1] | >300 | – |
| $H_2F$ | >150 | >50 | >20 |
| L-Sepiapterin | >150 | >150 | >150 |
| D-Neopterin | >150 | >50 | – |
| 7-Biopterin | – | >50 | – |

[1]No inhibition at the maximum concentration tested

contrast, their ability to utilise conjugated pterins was markedly different. Based upon the $GC_{50}$ values, folic acid would appear to be the most efficient in supporting growth of *T. brucei*. However, the maximum growth of these cells (cultures supplemented with between 10 nM and 500 µM folic acid) was only ~30% of the positive controls (Fig 5B). Likewise, $H_2F$ was only able to support the growth of *T. brucei* up to 80% of the maximum growth with growth inhibition evident above 50 µM (Table 2). Unlike, *C. fasciculata* pteroic acid was unable to support growth.

The pterin requirements for growth of *L. major* are in excellent agreement with the earlier work of Nare *et al.* [32] who reported L-biopterin, $H_2B$, L-neopterin and sepiapterin to be "good pterin nutrients" when cultures were supplemented with 5 µg ml⁻¹ (~20 µM) of these pterins. Consistent with our findings in Table 1, D-neopterin, 7-biopterin, pteroic acid, pteridoxamine, xanthopterin, isoxanthopterin and 6-carboxypterin were reported as "inactive pterin nutrients" in wild-type promastigotes. Unexpectedly, we note that reduced pterins ($H_2B$ and $H_4B$) had $GC_{50}$-values for *L. major* that were orders of magnitude lower than that for biopterin or neopterin, a feature that was not observed with *T. brucei* or *C. fasciculata* (Table 1). The reason for this striking preference in *L. major* for a reduced pterin is not clear. Perhaps uptake of biopterin

is less efficient than uptake of $H_2B$ or $H_4B$ (or its oxidation product $qH_2B$) in *L. major*, compared to *T. brucei* or *C. fasciculata*. Leishmania has fourteen members of the Folate Biopterin Transporter (FBT) family [52], some of which have been characterised [53,54]. In contrast, *T. brucei* has only seven [55], whereas *C. fasciculata* Cf C1 [56] has 51 putative FBT genes listed in TriTrypDB. Another explanation might be that biopterin (or neopterin) is less efficiently converted to their respective tetrahydro-forms in *L. major*. However, the comprehensive enzymatic studies on PTR1 in *L. major* [32] and *T. brucei* [2] would not support this. Moreover, it is difficult to envisage an explanation based on the presence or absence of qDPR activity, since this enzyme is present in *L. major* [33] and *C. fasciculata* [57], but absent in *T. brucei* [2].

*Further metabolism of pterins in C. fasciculata* - To determine whether pterins that support growth undergo further metabolism, parasites were cultured in the presence of a variety of pterins and/or folates (10 μM) and their intracellular pterins then analysed during log phase (Fig 6 and Table 3). No difference in intracellular pterin levels was observed in cells cultured with either $H_4B$ or L-biopterin, with $H_4B$ constituting over 96% of the total. The trace amount of pteridoxamine (0.6%) observed in $H_4B$ cultures is likely to be an extracellular breakdown product of this highly unstable pterin [51]. Interestingly, L-neopterin was found to be principally converted to tetrahydroneopterin ($H_4N$) in cells, but not further metabolised. As observed with cells cultured in L-biopterin or $H_4B$, over 98% of intracellular neopterin was present in the tetrahydro-form ($H_4N$), although total pterin levels within these cells were ~3.5-fold higher ($59\pm3$ μM).

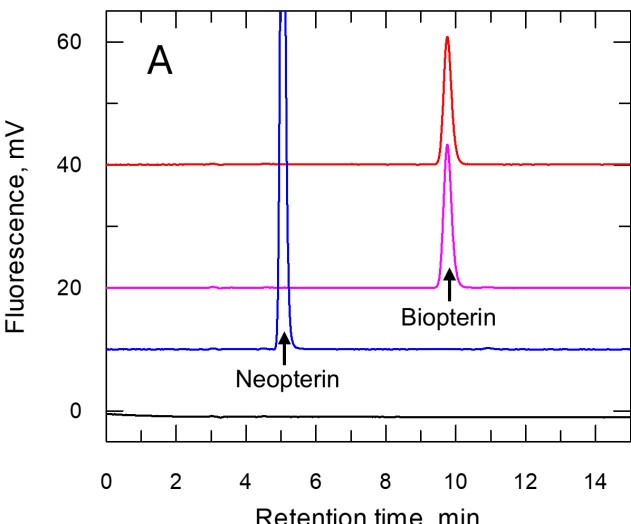
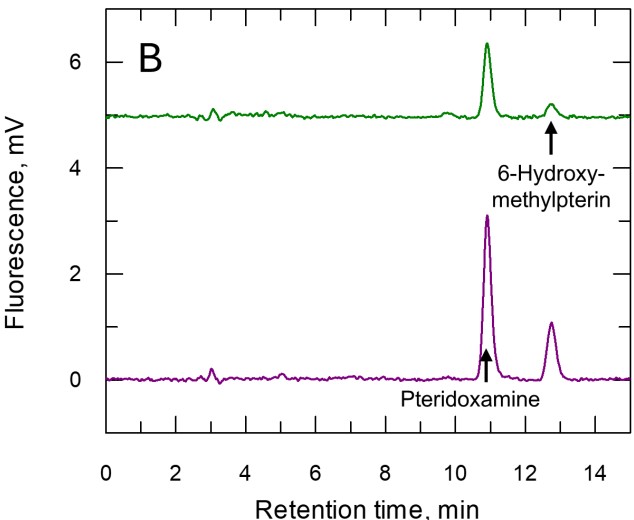

**Fig 6. Chromatograms illustrating HPLC-mediated separation of pterins.**

Table 3. Pterin content and interconversion in *C. fasciculata*. Parasites were cultured supplemented with 10 μM pterins and analysed for intracellular pterin content as described in the methods. Values are the means (μM) plus standard deviation of three biological replicates.

| Supplement | Biopterin | $H_4B$ | Neopterin | $H_4N$ | Pteridoxamine | HMP | $H_4$HMP |
|---|---|---|---|---|---|---|---|
| Biopterin | 0.71±0.05 | 16.9±0.2 | – | – | – | – | – |
| $H_4B$ | 0.51±0.04 | 15.9±0.1 | – | – | 0.095±0.01[a] | – | – |
| Neopterin | – | – | 0.77±0.10 | 58.6±3.1 | – | – | – |
| $H_2F$ | – | – | – | – | 2.8±0.02[b] | 0.14±0.01 | 0.89±0.02 |
| Folic acid | – | – | – | – | 1.4±0.05[b] | 0.068±0.003 | 0.009±0.004 |

[a]breakdown product of $H_4B$

[b]likely to be derived from tetrahydrofolates

Pterins were separated on a $C_{18}$ reverse phase column using a mobile phase of 20 mM sodium phosphate plus 4% methanol, pH 6.5. Fully oxidised pterins were detected fluorometrically using excitation wavelengths of 360 and 440 nm, respectively. Panel A: Representative chromatogram of extracts of *C. fasciculata* cultured in L-biopterin (red line), $H_4B$ (pink line), and neopterin (blue line), Panel B: folic acid (green line) and $H_2F$ (purple line), respectively. Note difference in scale in A and B. Quantification is provided in Table 3. Samples were prepared by oxidising cell suspension by iodine under acidic conditions. HPLC analyses were carried out on 10 µl of injected samples (equivalent ~1 x $10^6$ cells).

High concentrations of folate can support growth of *C. fasciculata* in the absence of pterins (Table 1). Analysis of *C. fasciculata* cultured under these conditions revealed that folate is recovered as pterins that co-elute with pteridoxamine and 6-hydroxymethylpterin (HMP) (Fig 6 and Table 3). Pteridoxamine is likely to be derived from intracellular tetrahydrofolates following acidic oxidation with iodine required for our analytical method [46], whereas HMP (predominantly present in the tetrahydro-form, $H_4HMP$) is a known metabolic product of folate [58]. Pteridoxamine and HMP were also present in *C. fasciculata* cells cultured in $H_2F$ with markedly higher levels of $H_4HMP$ plus HMP (~10-fold) compared to folate cultures. Unlike folic acid, reduced folates are highly unstable ($t^{1/2}$~5h) undergoing autooxidation at neutral pH in the absence of ant-oxidants to 6-formylpterin [59]. This might be taken up and reduced by PTR1 to dihydro-6-formylpterin which could then undergo reduction to dihydro-HMP by the action of a broad specificity aldehyde reductase [60]. Radiotracer experiments of Kidder and colleagues [58] identified biopterin, as well as HMP, as a metabolite of 2-[$^{14}C$]folate. Biopterin was not detected in our analyses, but it is worth noting that the amount of radiolabelled biopterin in the experiment by Kidder et al was only 0.25% of that of HMP, well below the limit of detection in our assay.

*Conclusion and future prospects.* This investigation has demonstrated that pterins are essential for growth in African trypanosomes as well as confirming and extending knowledge on the essential pterin requirements in *Crithidia* and *Leishmania*. A common feature of the pterins that supported growth in these organisms is the presence of at least one or more linear carbon substituents at position 6 of the pteridine ring with at least one hydroxyl group, ideally in the 1S configuration – cf. L-biopterin with, L- and D-neopterin (Table 1 and Fig 1). Further metabolism of oxidised pterins appears to be restricted to reduction to the tetrahydro-forms via PTR1, a potential drug target in *Leishmania spp.* [31,32,61] and *T. brucei* [2,34]. Which of these chemical features are critical for uptake, reduction to their active form or their vital biological function(s) requires further investigation. The metabolic functions of biopterin in other organisms are well established: principally involving hydroxylation of aromatic amino acids (phenylalanine, tyrosine and tryptophan) by their respective hydroxylases or hydroxylation and cleavage of glyceryl ethers by glyceryl ether monooxygenase. Although Beverley and colleagues have provided biochemical and genetic evidence for phenylalanine hydroxylase activity in leishmania [62], *Leishmania spp.* are auxotrophic for both phenylalanine and tyrosine [63–65], as are *C. fasciculata* [26] and *T. brucei* [66] that also lacks genes for amino acid hydroxylases [67]. These findings eliminate aromatic amino acid hydroxylation as the essential role for pterins in these trypanosomatids. All these kinetoplastids can synthesise and hydrolyse ether lipids [67–71]. However, the alkylglycerol monooxygenase activity in *L. major* uses NADPH rather than $H_4B$ as the cofactor used in other organisms [63]. Moreover, abolition of all ether lipid biosynthesis is not lethal in leishmania [72] or *T. brucei* [67].. A role as an anti-oxidant has been reported in leishmania [73,74], but it is difficult to envisage this as a specific and unique essential function given the plethora of other anti-oxidant defences in these organisms [75–77]. An overall summary of the metabolic pathways in these trypanosomatids (Fig 7) underlines the fact that, of the known biopterin-dependent pathways involving oxygen (phenylalanine, tyrosine, tryptophan hydroxylases, glyceryl ether monooxygenase and arginine-dependent nitric oxide synthase), only the non-essential phenylalanine hydroxylase is present in *Leishmania* and *Crithidia spp.* However, the fact that this metabolic cycle is retained suggests that another unidentified essential metabolite or essential biological process is involved.

One area for future research that has received little attention since the pioneering work of Kidder and colleagues is a potential role for pterins in the synthesis of lipids, specifically, unsaturated fatty acids and sterols. Growth requirement for pterin can be spared by the addition of crude lipid extracts from *C. fasciculata* [78,79]. Moreover, perturbations in lipid and

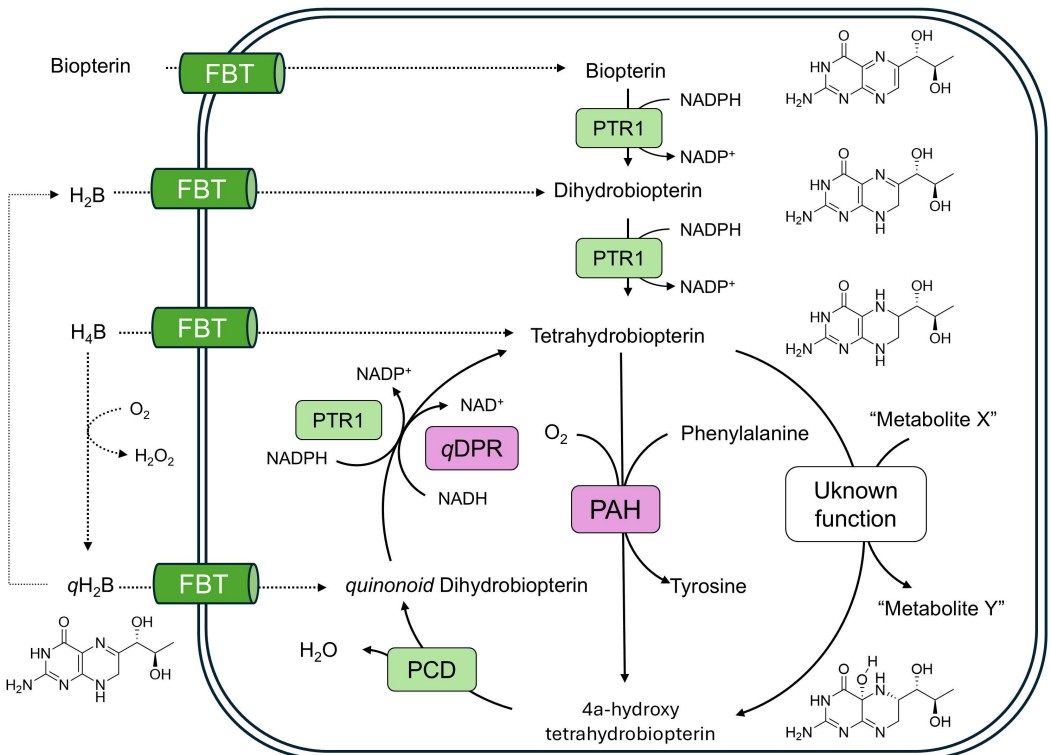

**Fig 7. Biopterin metabolic pathways in trypanosomatids.** Transporters and enzymes common to *L. major*, *T. brucei* and *C. fasciculata* are coloured in green; enzymes common to *L. major* and *C. fasciculata* are shown in pink. An unknown oxygen dependent essential metabolite "Y" is proposed to account for retention of the cyclical pathway. Abbreviations: FBT, folate biopterin transporter; PTR1, pterdine reductase; *q*DPR, quinonoid dihydropteridine reductase; PAH, phenylalanine hydroxylase; PCD, pterin carbinolamine dehydratase.

sterol metabolism of [$^{14}$C]-acetate have been reported in pterin-deficient parasites [80]. Our preliminary experiments with pterin-free subcellular fractions of *C. fasciculata* would support these earlier reports of growth sparing, suggesting this subject merits further investigation.

Seventy-five years have elapsed since the discovery of "Crithidia factor", yet we still do not understand the essential role(s) pterins in kinetoplastids. Further work in this area using the modern tools of metabolomics and lipidomics should resolve this unanswered question and could lead to the identification of novel targets for drug development.

## Supporting information

**S1 Table. Raw data used to build graphs in Figs 2A, 3, 4, 5A, 5B and 5C.**
(XLSX)

## Author contributions

**Conceptualization:** Alan H. Fairlamb.

**Data curation:** Alan H. Fairlamb.

**Formal analysis:** Alan H. Fairlamb.

**Investigation:** Han B. Ong, Susan Wyllie.

**Methodology:** Han B. Ong, Susan Wyllie.

**Supervision:** Susan Wyllie, Alan H. Fairlamb.

**Writing – original draft:** Han B. Ong.

**Writing – review & editing:** Han B. Ong, Susan Wyllie, Alan H. Fairlamb.

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
