## [Decision Letter · Decision Letter 0]

Dear Dr. Fairlamb,

Please submit your revised manuscript within 30 days Aug 23 2025 11:59PM. If you will need more time than this to complete your revisions, please reply to this message or contact the journal office at plosntds@plos.org. Please include the following items when submitting your revised manuscript:

Response to Reviewers
Revised Manuscript with Track Changes
Manuscript

Shaden Kamhawi

co-Editor-in-Chief

Paul Brindley

co-Editor-in-Chief

**Additional Editor Comments :**

**Journal Requirements:**

3) We note that your Data Availability Statement is currently as follows: "All relevant data are within the manuscript and its Supporting Information files". We noticed that there aren't any supporting information files uploaded in the online submission form.

4) Please provide a completed 'Competing Interests' statement, including any COIs declared by your co-authors. If you have no competing interests to declare, please state "The authors have declared that no competing interests exist". Otherwise please declare all competing interests beginning with the statement "I have read the journal's policy and the authors of this manuscript have the following competing interests:"

**Reviewers' comments:**

**Key Review Criteria Required for Acceptance?**

**Methods**

-Are the objectives of the study clearly articulated with a clear testable hypothesis stated?

-Is the study design appropriate to address the stated objectives?

-Is the population clearly described and appropriate for the hypothesis being tested?

-Is the sample size sufficient to ensure adequate power to address the hypothesis being tested?

-Were correct statistical analysis used to support conclusions?

-Are there concerns about ethical or regulatory requirements being met?

Reviewer #1: No issues

Reviewer #2: Figure 5: I have some concerns about using resazurin as an indicator of proliferation. Changes in cellular metabolic states may increase or decrease fluorescence in the assay, which does not necessarily correlate with cell proliferation. I suggest interpreting these data as cell viability rather than proliferation.

Reviewer #3: Yes, this is a well-conducted study to explore the requirements of pterin analogous for the growth of Crithidia fasciculata, Trypanosoma brucei and Leishmania major. The methods are appropriately described and support the conclusions.

**Results**

-Does the analysis presented match the analysis plan?

-Are the results clearly and completely presented?

-Are the figures (Tables, Images) of sufficient quality for clarity?

Reviewer #1: No issues

Reviewer #2: Yes

Reviewer #3: Yes, the results are clearly stated and followed the proposal to investigate the variability of pterin derivatives in supporting the growth of the three different species.

**Conclusions**

-Are the conclusions supported by the data presented?

-Are the limitations of analysis clearly described?

-Do the authors discuss how these data can be helpful to advance our understanding of the topic under study?

-Is public health relevance addressed?

Reviewer #1: see comments below

Reviewer #2: Yes

Reviewer #3: Yes, the conclusions confirms previous findings for the requirement of pterins in Crithida and Leishmania and extend it to T. brucei. It also demonstrated that preferential accumulation of transformed in hydroxylated forms (H4B) in T. brucei. Based on previous studies, the possible role of this compounds remained unknown.

**Editorial and Data Presentation Modifications?**

Reviewer #1: None

Reviewer #2: I believe the paper could be enriched by adding a schematic figure illustrating the metabolic pathways of pterin utilization in these parasites. This could broaden the work’s reach to other audiences.

Reviewer #3: The presentation is clear and I couldn't find possible required changes. A topic title could introduced before the ...In conclusion, (linde 388

**Summary and General Comments**

Reviewer #1: Ong et al. from the Fairlamb lab have studied pterin requirements for the growth of 3 different trypanosomatids. Pterins are known to be essential for the growth of these parasites although their exact function is still a mystery and differs from their role in other organisms. By using defined medium, they tested the growth of three parasite genus while complementing with a variety of pterin molecules. They have also studied the fate of the supplemented pterins by using HPLC. The fact that pterins are required for growth of these parasite is not new but by including a leishmania representative, T. brucei and Crithidia they were able to show some differences notably with Leishmania in comparison to the other genus (preference for reduced pterins and lack of a folate sparing effect). They also show that the absence of reduced pterins in stationary phase is most likely due to the exhaustion of the pterins in the culture medium. Overall this work was performed carefully and competently and few labs, if any, would have been able to make connections with the work of Kidder and Dewey, or Trager in the 50s-60s. I have a number relatively minor comments related to this work.

1. Since reduced pterins (BH4) is dominant intracellularly for all three parasites, I am wondering why T. brucei and Crithidia, in contrast to Leishmania, prefer oxidized pterins for growth. Does it have to do with their capacity to transport different form of pterins? Leishmania has has a large family of Folate-biopterin transporters (FBT) and maybe some of them specializes in the transport of BH2 and BH4. Does T. brucei and Crithidia have an extended FBT gene family?

2. This study has not really added to our vexing lack of understanding of pterin’s function in trypanosomatids (except their growth stimulating properties). The authors hypothesized that they may have a role to play in the synthesis of unsaturated fatty acids and sterols and they provide literature of the 60s to support this hypothesis. At lines 416-418, if I understand well, the authors make a statement that they may have evidence for this. Can the authors expand on this?

3. In table 2 what are the dash (-) indicating? Not done, no inhibition?

4. Title page: Why a hashtag sign for Ong?

Reviewer #2: Minor revision

Lines 213–216: I did not understand the purpose of comparing PTR1 activity with trypanothione reductase activity. What insight does this provide?

Figure 4A: The resolution is poor. Please replace it with higher-resolution images.

Figure 5: I have some concerns about using resazurin as an indicator of proliferation. Changes in cellular metabolic states may increase or decrease fluorescence in the assay, which does not necessarily correlate with cell proliferation. I suggest interpreting these data as cell viability rather than proliferation.

Lines 404–405: I would be cautious about generalizing for all trypanosomatids based on limited examples. Including T. cruzi in the discussion might allow such generalizations for pathogenic trypanosomatids.

I believe the paper could be enriched by adding a schematic figure illustrating the metabolic pathways of pterin utilization in these parasites. This could broaden the work’s reach to other audiences.

Reviewer #3: In this manuscript the authors re-explore in a comparative manner the requirements of pterin derivatives in the growth of three species of kinetoplastids. The study serves a premisse to further understand the role of these cofactors, essential in these organismos. However, the study advance quite little in the understanding of their functions.

PLOS authors have the option to publish the peer review history of their article (what does this mean? ). If published, this will include your full peer review and any attached files.

**Do you want your identity to be public for this peer review?** For information about this choice, including consent withdrawal, please see our Privacy Policy .

Reviewer #1: No

Reviewer #2: No

Reviewer #3: No

**Figure resubmission:****Reproducibility:** To enhance the reproducibility of your results, we recommend that authors of applicable studies deposit laboratory protocols in protocols.io, where a protocol can be assigned its own identifier (DOI) such that it can be cited independently in the future. Additionally, PLOS ONE offers an option to publish peer-reviewed clinical study protocols. Read more information on sharing protocols at https://plos.org/protocols?utm_medium=editorial-email&utm_source=authorletters&utm_campaign=protocols

---

## [Editor Report · Decision Letter 1]

Dear Professor Fairlamb,

We are pleased to inform you that your manuscript 'Comparative metabolism of conjugated and unconjugated pterins in Crithidia, Leishmania and African trypanosomes' has been provisionally accepted for publication in PLOS Neglected Tropical Diseases.

Best regards,

Martin Craig Taylor

Guest Editor

Hira Nakhasi

Section Editor

Shaden Kamhawi

co-Editor-in-Chief

Paul Brindley

co-Editor-in-Chief

This revision deals with all the reviewer's comments appropriately and is now acceptable for publication.

---

## [Editor Report · Acceptance letter]

Dear Professor Fairlamb,

We are delighted to inform you that your manuscript, "Comparative metabolism of conjugated and unconjugated pterins in Crithidia, Leishmania and African trypanosomes," has been formally accepted for publication in PLOS Neglected Tropical Diseases.

Best regards,

Shaden Kamhawi

co-Editor-in-Chief

Paul Brindley

co-Editor-in-Chief
